# BAT: Benchmark for Auto-bidding Task

## Abstract

The optimization of bidding strategies for online advertising slot auctions presents a critical challenge across numerous digital marketplaces. A significant obstacle to the development, evaluation, and refinement of real-time autobidding algorithms is the scarcity of comprehensive datasets and standardized benchmarks.

To address this deficiency, we present an auction benchmark encompassing the two most prevalent auction formats. We implement a series of robust baselines on a novel dataset, addressing the most salient Real-Time Bidding (RTB) problem domains: budget pacing uniformity and Cost Per Click (CPC) constraint optimization. This benchmark provides a user-friendly and intuitive framework for researchers and practitioners to develop and refine innovative autobidding algorithms, thereby facilitating advancements in the field of programmatic advertising.

## CCS Concepts

• **Applied computing → Online auctions**.

## Keywords

Online auctions, advertising platforms, RTB, autobidding

**ACM Reference Format:**
Anonymous Author(s). 2024. BAT: Benchmark for Auto-bidding Task. In *Proceedings of Make sure to enter the correct conference title from your rights confirmation emai (Conference acronym 'XX)*. ACM, New York, NY, USA, 9 pages. https://doi.org/XXXXXXX.XXXXXXX

## 1 Introduction

Modern online advertising systems enable the dynamic rendering of advertisements on web page in response to a user request [28]. The displayed advertisements are usually chosen from the available inventory according to specific criteria, such as relevance, temporal proximity, and performance metrics. These selected advertisements are then hierarchically organized in descending order based on the aforementioned criteria [35].

In the majority of instances, either all available advertising spaces or the most prominently positioned ones are allocated through an auction mechanism for each impression, competing among sufficiently relevant advertisements [31]. This setup is also known as sponsored search problem [10, 21]. The company owning the advertisement submits a bid, and advertising space is awarded to the highest bidder in the auction.

Given the immense scale of advertisements and the frequency of auctions occurring in real-time, manual bid setting becomes impractical [32], necessitating the development of optimal automated bidding algorithms and thereby setting the RTB problem [15]. The reliability and efficiency [9] of developed algorithms for such problem directly influence the effectiveness, targeting precision, and overall return on investment (ROI) in both advertising and trading domains.

Research on RTB algorithms is widely conducted for both VCG auctions ([26], [32], [4], [6], [2], [36]) and FP auctions ([11], [23], [24], [27], [25], [3], [14]), because these auctions have proven to be competitive [12].

Beyond the diversity in auction types, companies' objectives and constraints in advertising campaigns vary significantly. Budget constraints are ubiquitous, and companies often specify desired click volumes or maximum cost-per-click thresholds [15].

Developing and validating automated bidding algorithm is an essential step prior to production deployment. The scarcity of appropriate datasets for this task is a well-documented challenge in the field of automated bidding [23].

## 2 Contribution

In this study, we introduce BAT (Benchmark for Auto-bidding Task), a novel dataset designed to support the development and evaluation of automated bidding algorithms and related tasks. To enhance the dataset's usability and accessibility, we provide a detailed description of its contents. BAT comprises two distinct parts: data from 10 million FP auctions and 1 million VCG auctions, each collected over a month-long period on a major platform.

Moreover, we demonstrate the practical utility of BAT by introducing two innovative algorithms for RTB: Adaptive Linear Model (ALM) and Traffic-aware PID (TA-PID). These algorithms have demonstrated their effectiveness in production environments, are straightforward to implement, and serve as a solid foundation for the development of more sophisticated methods.

Furthermore, we enhanced the automated bidding algorithm M-PID from [34] by leveraging specific dataset fields, resulting in a significant performance improvement compared to the baseline. We also included two additional baselines: the budget pacing system Mystique [29] and an autobidding algorithm with budget and ROI constraints [19].

To ensure reproducibility and facilitate further research, we make the source code used in our analysis publicly available. This code serves as a practical guide for interacting with BAT and provides a solid foundation for researchers and practitioners to build upon our work.

## 3 Related works

Development of RTB algorithms is critically important for both individual users and large advertisers, as well as for auction platform owners, to investigate the optimality, fairness, and efficiency of auction processes. The performance of autobidding algorithms under

 

real-world conditions can be evaluated using specialized datasets that include logs from various auction types across a diverse set of companies from different information domains [23]. This is crucial because advertising characteristics for products like automobiles and food differ significantly in terms of temporal periodicity, click-through conversion rates, and other metrics. These datasets should encompass key company and advertisement indicators, assess the click probability of winning ads, and provide statistics on auctions won in simulation, expenditures, clicks, and conversions for the algorithm under test [37].

In the context of this task, the 2014 IPinYou dataset [17]) remains one of the most pertinent resource available. Developed for the KDD Cup competition on RTB, this dataset encompasses ad features and bid prices, with the target variable (winning price or cost-per-click) to be predicted, alongside a substantial volume of logged auctions. Moreover, it incorporates contextual features pertaining to user interests and ad slot parameters. Comprising several million bid requests, the IPinYou dataset is conducive to robust statistical analysis and machine learning model training. However, it is limited to 9 advertisers, each representing a distinct logical category, which poses challenges in simulating competition among a more extensive array of ads driven by specific algorithms. Furthermore, the dataset's collection in 2013 may impact the relevance of its content to contemporary trends and technologies in online advertising. For instance, the dataset exclusively represents second-price auctions, which presents limitations for many modern platforms that have adopted alternative auction mechanisms [23].

In 2024, a dataset from Alibaba was released [33], developed for testing RL algorithms for solving the RTB problem with a Cost per Action (CPA) constraint. It contains a Generalized Second Price auction with 170 million records, the number of advertisers is 48. This dataset contains the winning price and the bids made in each auction, as well as the conversion action probability. Auctions are being implemented for 3 slots. The dataset contains the key components of an RL problem: states, actions, rewards, and environmental dynamics, making it ideal for training RL algorithms in the context of online advertising.

Since, as far as the authors are concerned, these two datasets are the only open datasets in the field of autobidding task, the algorithms are most frequently tested on private, closed datasets, which can be attributed to the need for anonymity on proprietary platforms [23].

A large number (more than 9000) of advertisers in the BAT dataset, participating in more than 10 million auctions, provides the opportunity to test algorithms representatively. Aggregating data on CTR and CVR for ads is an advantage of the dataset; it expands the possibilities of implementing a wide range of algorithms using these values. The data is presented for two types of auctions - VCG and FP, which in the context of the modern large-scale transition of platforms from second-price to first-price auctions [13] is of undoubted interest for the scientific community from the perspective of developing and testing algorithms in various formulations of the problem.

In the BAT dataset, the auction results contain a wide range of predictions for click and conversion events, the increase in visibility for the ad and statistics on budget write-offs in connection with clicks on the ad (see below in section Dataset description). Auctions

are implemented for all slots, which provides wide variability in display outcomes, which is important to consider when participating in auctions on many platforms. The format of the dataset fields is as close as possible to the data used in production on large advertising platforms, suggesting the use of computationally simple in interference and effective algorithms for autobidding.

In addition to the dataset we present several RTB baselines. Complex and efficient algorithms dedicated to the task of budget pacing are constantly being developed [5], [20]. We will use in our baselines principles, which are commonly used for this type of problem in applications on modern platforms due to their efficiency [34], [16], [7].

## 4 Dataset description

Let us provide a comprehensive description of our datasets. Each dataset (VCG and FP) comprises three distinct components: (a) campaigns and their permanent attributes, (b) auction outcomes, and (c) traffic data.

### 4.1 Campaigns

This component of the dataset encompasses information pertaining to the invariant parameters of the participating advertising campaigns (see Table 1).

| loc_id | 653248 | 630730 |
| --- | --- | --- |
| campaign_id | 272505312 | 271449978 |
| item_id | 3660681800 | 2561215400 |
| campaign_start_date | 1970-01-27 | 1970-01-27 |
| campaign_end_date | 1970-02-03 | 1970-02-03 |
| campaign_start | 2302355 | 2253120 |
| campaign_end | 2907155 | 2857920 |
| auction_budget | 378227125476 | 4282490290176 |
| microcat_ext | 4928 | 4147 |
| logical_category | 2.33 | 3.23 |
| region_id | 653420 | 630660 |
| platform_p | [0.5 0. 0.25 0.25] | [0.24 0.08 0.24 0.44] |

Table 1: Campaigns statistics data format.

- loc_id - unique identifier for the location where a transaction can be made to purchase the object advertised in the campaign,
- campaign_id, item_id - unique identifier assigned to each promoted advertising campaign and its corresponding item, respectively,
- campaign_start_date, campaign_end_date - starting and ending dates of the campaign, which have been shifted to maintain anonymity,
- campaign_start, campaign_end - unix-like timestamps representing the starting and ending times of the promotional campaign,
- auction_budget - total monetary budget allocated to each campaign,
- microcat_ext - identifier for the micro-category to which the advertised item belongs,

- logical_category (categorical variable) - reference index indicating the item's category. This index consists of two parts separated by a dot: the global logical category index and the subcategory index,
- region_id - reference index representing the geographical location of the item, providing a broader spatial context compared to the loc_id.

Furthermore, data regarding the frequency of advertisement participation in auctions across the four user-accessible platforms is available. This information, currently not utilized in the algorithms, may prove valuable for future development of complex autobidding algorithms by employing this field as a categorical feature.

## 4.2 Auction statistics

The data presented in Table 2 contains auction statistics for campaigns from Campaigns part. We propose that this dataset component be utilized in auction simulations, kept separate from the actual bidding method, and provided solely for training purposes.

| item_id | 3315908300 | 3315908300 |
|---|---|---|
| campaign_id | 231571725 | 231571725 |
| period | 784791.0 | 791991.0 |
| contact_price_bin | 245 | 240 |
| AuctionVisibilitySurplus | 0.771 | 0.348 |
| AuctionClicksSurplus | 0.451 | 0.405 |
| AuctionContactsSurplus | 0.212 | 0.205 |
| AuctionWinBidSurplus | 725.661 | 288.975 |
| CTRPredicts | 0.0 | 0.0 |
| CRPredicts | 0.0 | 0.0 |
| AuctionCount | 2.0 | – |

**Table 2: Auction statistics data format.**

Each log represents one participation of an advertising campaign in one auction.

- campaign_id, item_id - same as mentioned above,
- period - timestamp for which the auction data was aggregated,
- contact_price_bin - discreteprice bin value, which can be mapped to the actual auction bid using a logarithmic transformation function - $\gamma^{bin}$ (in our case $\gamma$ = 1.2),
- AuctionWinBidSurplus, AuctionVisibilitySurplus - expected incremental cash write-offs for auctions at the current bid level relative to the previous auction position, and the mathematical expectation of additional visibility gained with the current bid. Visibility is defined as the probability that a user will scroll down and view the advertisement, with a maximum value of 1. "Incremental" refers to the difference in visibility between the current bin and the previous bin (smaller by 1),
- AuctionClicksSurplus, AuctionContactsSurplus - expected increase in user clicks and contacts within the specified time frame compared to participating in the auction with the previous bin,
- CRPredicts, CTRPredicts - values of item CTR and CR aggregated by item features,

- AuctionCount (for VCG auctions) - number of observed auctions from which the data was aggregated.

## 4.3 Traffic

The Traffic component od dataset (see the Table 3) consists of the information about a contacts-over-time distribution on the auction platform. This data describes how the traffic is spreading over week for each region separately.

| region id | dow | hour | traffic share |
|---|---|---|---|
| 645530 | 1 | 0 | 0.001704 |
| 645530 | 1 | 1 | 0.000917 |
| 645530 | 1 | 2 | 0.000546 |
| 645530 | 1 | 3 | 0.000314 |

**Table 3: Traffic share data format.**

- region_id - identifier of region, similar to other dataset components;
- dow - number of day from 1 to 7, where numeration starts from Sunday;
- hour - hour of collected statistics, from 0 to 23;
- share - portion of total contacts during the week. The sum of all traffic in the region for a week is 1.

This information can be used to specialize the algorithm regarding different types of traffic shares and be used as a prediction when setting a bid.

## 5 Dataset collection, filtering and sampling

The VCG auction data was collected in March/April 2024, FP auction data - in July/August 2024.

Data acquisition was performed by utilizing auction results from search and real advertising campaigns participating in these auctions. After determining the outcomes of each company's participation in the auctions, all results for campaigns (including winning or losing at auction, corresponding bid, CTR, CR, and resulting position in the auction) were aggregated based on price bins and time periods (with 1-hour discretization).

The surplus fields were derived by calculating the difference between the sum of the relevant parameters (such as clicks, visibility, and others) for the current price bin and the corresponding sum for previous price bin. The calculation took into account the resulting position and its visibility. CTR and CR were estimated as the average of the corresponding values but exclusively for the represented campaign. The number of aggregated auctions was recorded in the AuctionCount field.

Statistics with VCG auction were collected using 10% of auction results, logged to form one period of statistical data for each campaign. These datasets contain information about 2500 sampled campaigns with statitics over 21 days period.

Statistics with FP auction was formed by logging 100% of auctions data but only for 1% of advertising campaigns. We consider this dataset to have more descriptive power over VCG auctions dataset. This dataset contain information about 9000 sampled campaigns with statistics over 21 days period.

Furthermore, the datasets were filtered to contain only campaigns that are fully covered by statistics data over the lifetime of the campaign, and the auction data was logged correctly. The filtering process included several steps: removal of invalid or incomplete data points, ensuring all necessary metrics were present and positive, verification of data completeness, including checks for full period coverage and alignment of log start times with campaign start times, exclusion of campaigns shorter than one day or with duplicate entries, other various quality indicators, such as budget adequacy for both VCG and FP auctions, sufficient click and contact volumes, and minimum campaign durations.

## 6 Dataset statistics

Logging and aggregating a large amount and range of data on advertisements and auctions allows you to use the dataset for debugging RTB algorithms with fine-tuning on many parameters. Here we provide a brief overview of the details of the statistics.

The weekly traffic distribution graph (see Fig. 1) indicates significant daily fluctuations - a 30-fold difference with a maximum at noon and a minimum at 3 a.m., which should be taken into account when calculating rate changes during the day. The maximum activity in all regions is observed on Monday, the minimum on Saturday.

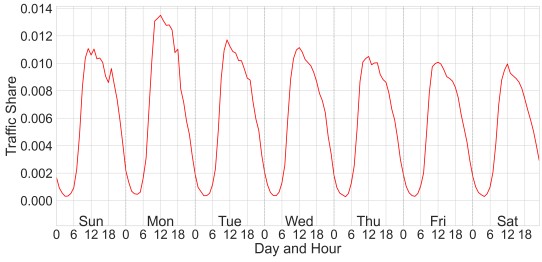

**Figure 1: The statistics on week traffic distribution for all regions on average.**

Table 4 shows the distribution of logged campaigns by timelife: most campaigns participated in auctions for 1 day, the next position is duration for a week, the remaining campaigns make a smaller contribution. These statistics are explained by the naturalness and convenience of choosing a reporting period of 1 day and 1 week.

| Days | VCG | | FP | |
|---|---|---|---|---|
| | Campaigns | % | Campaigns | % |
| 1 | 24280 | 75.85% | 308732 | 83.46% |
| 2-6 | 2205 | 8.45% | 27433 | 7.41% |
| 7 | 4985 | 15.57% | 30588 | 8.27% |
| 8-14 | 41 | 0.13% | 3180 | 0.86% |

**Table 4: Distribution of campaigns by timelife.**

Figure 2 shows the dependence of the average increase in write-offs, visibility, clicks and contacts on the bin for one random microcategory for both auction types, compared to previous bin. These dependencies are extremely useful for analyzing and adjusting the

operation of algorithms, for example, one can estimate that for contact price bin 55 for VCG auctions and 40 for FP auctions, there is a maximum increase in visibility and budget write-offs, as well as a maximum increase in clicks and contacts. The characteristic growth of surpluses at small bins is the result of the reserve prices using.

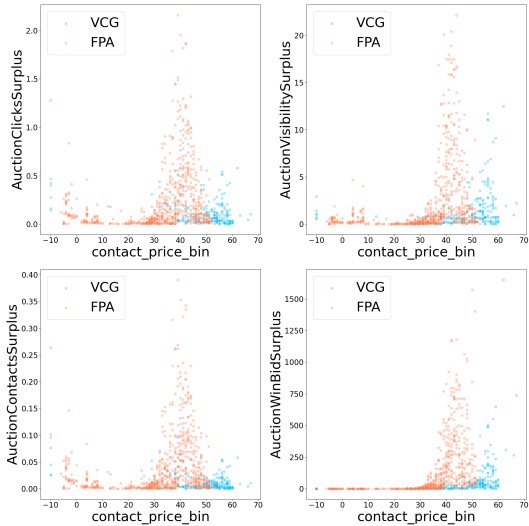

**Figure 2: Dependences of AuctionClicksSurplus, AuctionVisibilitySurplus, AuctionContactsSurplus, AuctionWinBidSurplus on contact price bin for an example of micricategory.**

On average, auction items are more expensive at night, as shown in Fig. 3. This is due to the decrease in traffic during these hours. However, this dependence has a large scatter, as demonstrated by the curves of several specific campaigns.

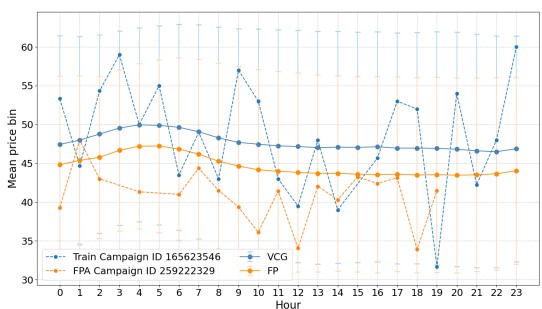

**Figure 3: Dependence of contact price bin on hour: blue shades for VCG, orange for FP. The graph shows the time-averaged price bin, standard deviation, and one example of values for a specific campaign for each auction type.**

Table 5 provides a concise descriptive analysis of campaign budgets in both components (VCG and FP) of our dataset. This distribution is particularly valuable for consideration when designing and analyzing simulations. For instance, given our constraints, it

significantly influences the theoretical maximum number of clicks that campaigns can procure at a fixed CPC.

| Budget | VCG | | FP | |
|---|---|---|---|---|
| | Campaigns | % | Campaigns | % |
| 0-500 | 6349 | 73.53 | 0 | 0.00 |
| 500-1000 | 635 | 7.35 | 0 | 0.00 |
| 1000-10000 | 1233 | 14.28 | 6895 | 82.96 |
| 10000-50000 | 418 | 4.84 | 1416 | 17.04 |

**Table 5: Distribution of campaigns by initial budget.**

The characteristics of auction prices and their quantity may have a strong dependence on the logical category. Figure 4 presents a histogram illustrating the distribution of campaign volumes across various logocal categories. The category nomenclature employs separated by a dot hierarchical subdivision, so for example 1.2 could be the equivalent of Cars.Ferrari.

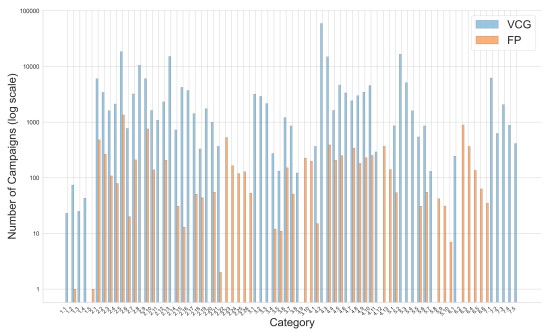

**Figure 4: The number of campaigns per logical category.**

Figure 5 illustrates the relationship between the winning bid and the predicted CTR for a single micro-category. While the observed relationship is non-monotonic, there is a discernible trend indicating an increase in the average bin value for higher CTRs.

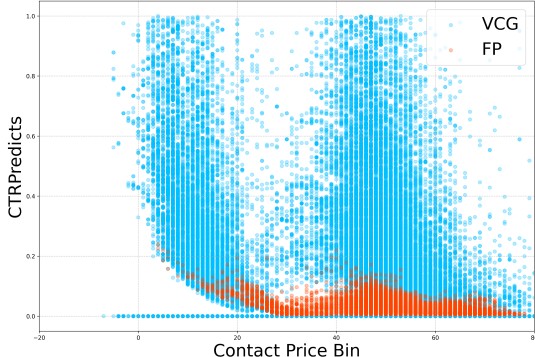

**Figure 5: Dependence of CTR on contact price bin for two auction types.**

# 7 Problem formulation

The task is based on a common problem, which is caused by the need for modern platforms to provide advertisers with an automated bidding strategy for their advertising campaigns with a limited budget. The goal is to have a fixed duration during which spending is desirable to be consistent, and the number of additional clicks received during the campaign should be as high as possible.

This setting coincide with the work of [34], which considered the case where each ad campaign occurs in $N$ auctions per day. Each auction has winning price $wp_i$, which is determined when platform gets all bids. In VCG auction, if $bid_i$ is higher than $wp_i$, then campaign will win that auction, platform sets $x_i$ to be 1, and 0 otherwise. In FP case, if campaign wins the auction the $bid_i$ is equal to $wp_i$. Budget of the agent is decreased on $bid_i$ if user clicks on the ad.

Authors of work [34] solve the task under constraints of budget $B$ and average CPC which has to be less than $C$. In the original work the problem was formulated as:

$$\max_{x_i} \sum_{i=1\ldots N} x_i \cdot CTR_i \cdot CVR_i$$
$$\text{s.t.} \sum_{i=1\ldots N} x_i \cdot wp_i \leq B$$
$$\frac{\sum_{i=1\ldots N} x_i \cdot wp_i}{\sum_{i=1\ldots N} x_i \cdot CTR_i} \leq C$$
$$\text{where} \quad 0 \leq x_i \leq 1, \forall i$$

Since we aim to address a similar task, we will utilize this problem formulation. The budget constraints for the campaigns ($B$) are taken from Campaigns data, 'auction_budget' field, while the cost per click ($C$) constraints are set manually depending on the experiment and are uniform across all campaigns. The winning price ($wp$) is determined based on aggregated data for each individual campaign, specifically from the 'contact_price_bin'.

The problem under consideration is a linear programming problem. Authors turn to the primal-dual method, as described in [34], and obtain the known theorem for the optimal bid, which relies on the solution of the dual problem. We use this algorithm as one of baselines, introducing into the formula a dependence on traffic distribution. We also use algorithm from work [19] which takes into account the CPC constraint mentioned above.

We also suggest testing two algorithms that have proven empirically successful in the budget pacing process, as they are already in use in the production environment. These algorithms, as well as [29], aim at uniformity of spending, which in real bidding is necessary to avoid the situation of buying clicks too quickly at too high price. These algorithms do not have a cost-per-click limit.

We will compare these algorithms to explore their effectiveness in several disciplines: budget pacing, satisfying the CPC condition, and buyout of the largest number of clicks per budget.

# 8 Baselines

We have tested our dataset on several baselines, taking into account the specifics of the data we provide.

We introduce the terms of traffic share as shown in the Figure 6. The time interval between $t_{k-1}$ and $t_k$ represents the time window that corresponds to a single step of the algorithm. $T_k$ and $T_{k-1}$

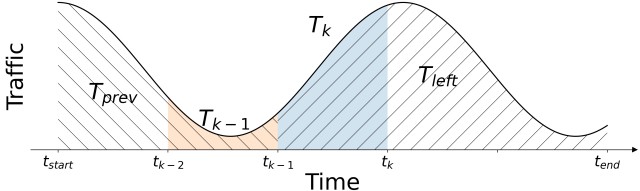

**Figure 6: Traffic definitions.**

are denoting expected traffic share in the current and previous auction time windows; $T_{left}$ and $T_{prev}$ are signifying remaining traffic share and traffic share from start up to now for the campaign. Additionally we use notation $T_{all} = T_{prev} + T_{left}$.

Consequently, the bid is discretized as $\Delta = \lfloor \frac{\log(b)}{\log \gamma} \rfloor$. Since, to compute a bid, the algorithm uses formula $b = \gamma^{\Delta}$, with the parameter $\gamma$ is confined by the design of the auction system. We will name $B_0$ as campaign's initial balance, and $B_k$ as campaign's balance at the moment $k$.

## 8.1 Adaptive Linear Model (ALM)

We use a fast and easy-to-implement baseline of an autobidding system employing linear prediction techniques, drawing inspiration from existing linear models utilized for bidding prediction [8], [22], [30], [32].

In the event of absence of prior bidding data for the campaign, the algorithm would initiate with an initial bid $b_0$, as each campaign in the dataset undergoes a cold start. Therefore, we selected a value for $b_0$ at which this problem occurred least frequently for all campaigns, iterating over it as a hyperparameter with optimization of the baseline metrics.

By considering $\hat{B}_n = \frac{B_n}{B_0}$ as the relative current campaign budget and $\hat{B}_{n-1} = \frac{B_{n-1}}{B_0}$ as the relative previous balance, with a slope

$$k = \frac{\hat{B}_n - \hat{B}_{n-1}}{T_n - T_{n-1}},$$

the algorithm conducts a linear prediction of the campaign budget output at the end of campaign lifetime:

$$\hat{B}_{left} = \hat{B}_n + k \cdot T_{left}.$$

---
**Algorithm 1** ALM
---
**Input**: Campaign, campaign's budget $B$, expected traffic distribution $T_n, 1 \le n \le N$
**Parameters**: Degree base $\gamma$, control parameter $\beta$, cold start value $b_0$
**Output**: Bids for each timestamp
1: Play bid $b = b_0$, $\Delta_0 = \log_\gamma b_0$
2: **for** $n$ in $1, ..., N$ **do**
3:     Receive clicks, spend budget
4:     Compute relative budget $\hat{B}_n, \hat{B}_{n-1}$ and slope $k$
5:     Compute $\hat{B}_{left}$ and then bin $\Delta_n$
6:     Update and play bid $b \leftarrow \gamma^{\Delta_n}$
7: **end for**
---

Subsequently, it calculates $\Delta_{n-1}$, where the current bin is $\Delta_n = \Delta_{n-1} + B_{left} \cdot \beta$, with control parameter $\beta$ chosen empirically.

In the final stage, the algorithm clips relative bin change $\Delta_n$ to avoid excessive fluctuation with clip boundaries (also as hyperparameters), and then computes $b_n$ based on the calculated bin.

## 8.2 Traffic-aware PID (TA-PID)

The classic PID uses the difference between the true and estimated value, to produce a control signal. This control signal is then sent to adjust the system's input.

Since PID controllers are still used as baseline methods [18] and in industries [5], we decided to propose this method as one of the baseline methods to make our dataset more accesible to use.

Keeping the previous designations of variables the same, we will describe a baseline based on a PID controller for managing a bid by comparing the spending speed with a reference value, taking into account historical traffic data.

As in the previous baseline, the algorithm would initiate with fixed bid $b_0$ for all campaigns.

Then algorithm begins by calculating the desired average budget spending rate $s_{ideal}$:

$$s_{ideal} = \frac{B_0}{T_{all}}$$

Then we calculate the control error as the difference between the desired and actual spend rates:

$$e_n = s_{ideal} - s_n = s_{ideal} - \frac{B_0 - B_n}{T_{prev}}$$

The PID controller takes $e_n$ to calculate the exponent bin adjustment $\Delta$ with the coefficients of proportional $k_p$, integral $k_i$ and differential $k_d$ dependence:

$$u(n) = k_p e_n + k_i \sum_{t=1}^{n} e_t (T_t - T_{t-1}) + k_d \frac{e_n - e_{n-1}}{T_n - T_{n-1}}$$

$$\Delta_n = \Delta_{n-1} + u(n)$$

.

---
**Algorithm 2** TA-PID
---
**Input**: Campaign, campaign's budget $B$, expected traffic distribution $T_n, 1 \le n \le N$
**Parameters**: Degree base $\gamma$, PID coefficients $k_p, k_i, k_d$, cold start value $b_0$
**Output**: Bids for each timestamp
1: Compute desired average budget spending rate $s_{ideal}$ and
2: Play bid $b = b_0$
3: **for** $n$ in $1, ..., N$ **do**
4:     Receive clicks, spend budget
5:     Compute control error $e_n$ and control signal $u(n)$
6:     Update $\Delta_n$ by $u(n)$
7:     Play bid $b \leftarrow \gamma^{\Delta_n}$
8: **end for**
---

## 8.3 Model predictive PID (M-PID)

This baseline was described in [34]. The authors use formula for optimal bid:

$$\text{bid}_n = \frac{1}{p+q} \cdot CTR_n \cdot CVR_n + \frac{q}{p+q} \cdot CTR_n \cdot C$$

where $p$ and $q$ are correspond to budget spending and CPC. These parameters will be used as reference signals for PID. M-PID also involves taking into account the indirect influence of reference signals on each other.

We modify the authors version of the PID for the described task as follows.

Our goal is to help advertisers to maximize the quantity of conversions with the budget $B_0$, get desired total number of clicks $Clicks$, and spend the budget as evenly as possible over a given period campaign lifetime $Time$. For the CPC reference, we use the ratio

$$CPC = \frac{B_0}{Clicks}.$$

To determine the budget spend reference for each step, we also normalize the current campaign's balance relative to the remaining traffic aggregated for the campaign's region, ensuring uniform ideal spending $s_n$ at the moment $n$:

$$s_n = B_n \cdot \frac{T_{cur}}{T_{left}},$$

The weights $k_p$, $k_i$, and $k_d$ are determined during offline testing and adjusted during online testing. Otherwise the same formulas and algorithm for PID as in [34] are applied.

## 8.4 Mystique

The algorithm in [29] optimizes the linearity of budget spending based on the expected lifetime of the campaign and the total budget. If the campaign experiences underspending or overspending in relation to the linear function, the algorithm, based on the difference between the desired and actual spending, as well as the slope of the desired and actual spend curves, changes the rate to reduce this difference.

Note that we implement only bid control from the algorithm, without implementing a daily update of the desired spend curve, for comparability of the algorithm's work with the work of M-PID.

## 8.5 BROI (Budget-ROI)

This algorithm has been selected due to its robust theoretical foundations and practical applicability, as well as its consideration of agents utilizing an equal bidding strategy, which is essential for the platform in question.

In the subsequent section, the optimistic variant of the algorithm proposed by Lucier et al. will be adopted. This study introduced an autobidding algorithm that integrates budget and return on investment (ROI) constraints. For the purposes of this analysis, ROI is interpreted as a cost-per-click (CPC) constraint. A significant theoretical finding of this research indicates that if all participants in the auction employ this algorithm, the resulting liquid welfare across all rounds can achieve at least fifty percent of the expected optimal liquid welfare. This algorithm has been selected due to its theoretical foundations and practical applicability, as well as its consideration of agents using an equal bidding strategy, which may be essential for some platforms.

## 9 Experiments and Metrics

This section will present experiments and relevant metrics to examine how the constraints for adaptive budget pacing and CPC tasks influence the performance of the provided algorithms. This analysis will focus on constraints beyond the budget constraint, which must always be satisfied.

## 9.1 Budget pacing experiment

The first 4 baselines (Linear, TA-PID, M-PID, Mystique) were developed for budget pacing, so the first experiment will be conducted for them with minimization of the deviation of the spend function from the uniform, normalized to traffic share.

To ensure uniformity in spending, a metric $RMSE_T$ is proposed and used to optimize baselines hyperparameters. This metric could be measured as the RMSE between the user's actual spend and an ideal spending curve, normalized by traffic share. This is calculated as follows: normalize hourly traffic for the campaign's lifetime $\hat{T}_n = \frac{T_n}{T_{all}}$, calculate the ideal budget at time $n$: $B_n^* = \hat{T}_n \cdot B_0$, and compute RMSE: $RMSE_T = \sqrt{\frac{\sum_{n=1}^{N}(B_n^* - B_n)^2}{N}}$, where $N$ is the number of time points.

Since M-PID has an additional constraint on the CPC, which is set equal to the initial budget so that the contribution of this condition does not affect the bidding results.

The Sum Click Ratio (SCR) will also be measured - sum of clicks achieved for all campaigns in experiment to compare the efficiency of algorithms.

## 9.2 CPC constraint experiment

The second experiment will be held for algorithms optimizing the solution of the RTB problem with the constraint of the CPC: M-PID and BROI. The CPC will be set 10 times smaller than the average value for the logical category under study, in order to formulate a result that is obviously difficult to achieve.

The metrics under study (used to optimize hyperparameters) will be Relative Cost Per Click, $REL\_CPC = CPC_{Real}/CPC$, where $CPC_{Real}$ is empiric mean cost per click for all campaigns.

## 9.3 Click sum maximizing experiment

Also, having equalized the chances of two types of algorithms (with and without CPC constraint) as in experiment 1, setting CPC equal to the initial budget, we examine the SCR metric itself to directly evaluate the efficiency of algorithms without CPC constraints.

## 10 Experimental settings

This section provides a detailed overview of the auction simulation process for all baseline models.

At the beginning of the simulation, we accept campaigns from both parts (VCG and FP) of the BAT as input, setting the budget and campaign lifetime based on statistics.

The simulation then commences. At each timestamp (one hour), the algorithms determine the bid for that period. The campaign budget for each timestamp is reduced by expected price of auction

participation. For the FP auction the expected price is the product of AuctionContactsSurplus and bid as if we buy each auction with defined bid. For the VCG auction, the result price is the sum of AuctionWinBidSurplus for each bin less or equal than campaign's one. Each campaign receives feedback at every timestamp, including write-off, additional clicks, contacts, visibility, and the number of won auctions.

It's important to note that the mechanism uses reference CTR and CVR values, which are necessary for solving the Linear Programming problem in M-PID, as mentioned in the original article [34], and are also part of the solution in BROI [19]. For estimating CTR and CVR, we utilize aggregated statistics from the Auction Statistics dataset, compiling data hourly, by category, and current bid range, since both CTR and CVR tend to strongly increase with bid value.

To optimize parameters more effectively, the campaign dataset is divided into two subsets, $S_1$ and $S_2$, ensuring that all campaigns in $S_1$ conclude before any campaign in the validation set begins. A Bayesian optimization package Optuna [1] is utilized on $S_1$ to maximize the metric involved, calculated as the total across all campaigns and then averaged. This method helps identify the optimal values for bidder constants for all baselines. Finally, the bidder mechanism is evaluated on $S_2$ to measure how effectively our parameter optimization performs, offering a more reliable assessment than optimizing on the entire dataset.

Note that the benchmark implements a convenient feature that allows for the generation of detailed plots, enabling users to observe the behavior of auction bids, the dynamics of budget expenditure, the number of clicks won, the average price per click won, and the speed of spending for each timestamp. In addition, a visualization of the comparison of the preformance of each of the prescribed baselines on the selected campaign instance is provided. This option makes debugging the algorithm and determining its behavior in edge cases convenient and user-friendly.

## 11 Experimental results

This section provides a quantitative analysis of the baseline performance based on the metrics, with results aggregated for logical category 1.

| | $RMSE_T^{VCG}$ | $SCR^{VCG}$ | $RMSE_T^{FP}$ | $SCR^{FP}$ |
|---|---|---|---|---|
| ALM | 8.73 | 756127 | 1.27 | 1612707 |
| TA-PID | 1.42 | 896534 | 1.61 | 1245836 |
| M-PID | 1.38 | 917521 | 1.26 | 1235585 |
| Mystique | 1.25 | 822787 | 1.18 | 696963 |

Table 6: The results of experiment 1 - tuning budget pacing.

Table 6 displays the results of first experiment focused on achieving the most uniform spending. The leading algorithm, Mystique, performs best in both auction types, as this was the target metric in its development. M-PID takes second place for FP auctions, while TA-PID comes in second for VCG auctions. ALM was the least effective in achieving uniform spending for VCG auctions, but its results for FP auctions are close to the winning algorithm

| | $REL\_CPC^{VCG}$ | $REL\_CPC^{FP}$ |
|---|---|---|
| M-PID | 0.91 | 0.49 |
| BROI | 0.88 | 0.94 |

Table 7: The results of experiment 2 - tuning CPC.

Table 7 displays the results from second experiment focused on satisfying the CPC constraint. It is evident that the algorithms perform with varying success in FP and VCG auctions. Notably, M-PID demonstrates significantly better results in FP auctions compared to its competitors than it does in VCG auctions.

| | $SCR^{VCG}$ | $SCR^{FP}$ |
|---|---|---|
| ALM | 662466 | 1085836 |
| TA-PID | 909282 | 1478538 |
| M-PID | 889251 | 1240244 |
| Mystique | 932152 | 1073291 |
| BROI | 495169 | 1098184 |

Table 8: The results of experiment 3 - tuning sum of clicks.

Table 8 shows the results of third experiment focused on maximizing click gains. The leading algorithms, TA-PID and ALM, achieve very similar results. Additionally, M-PID demonstrates excellent performance in FP auctions, while Mystique achieves strong results in VCG auctions.

In addition to the high efficiency of the mentioned algorithms in achieving the desired outcomes, we would like to highlight the transparency of the autobidder algorithms. This clarity allows researchers to easily track the key features of the algorithm's behavior—specifically, based on traffic in our case, and on other relevant parameters in general. Furthermore, the consistent performance of these algorithms across the dataset validates its effectiveness as a benchmark for autobidding research

## 12 Conclusion

The work presents a user-friendly benchmark BAT for RTB algorithms developement. The benchmark includes a new large-scale dataset, containing data on both VCG and FP auctions. This dataset reveals detailed information not only about the distribution of winning bids but also about traffic details, statistics of logical categories and geographic regions, aggregated information about CTR, CVR ads, purchased clicks, visibility, contacts, and funds deducted when winning an auction. Additionally, the dataset includes logs of all auction winners and several losers to ensure the completeness of statistics.

In addition, a series of RTB algorithms (novel and well-known) was implemented within the benchmark. Metrics were proposed, and experiments were conducted for various formulations of the budget pacing problem. This makes the technique of working with the dataset as transparent as possible. We believe a benchmark based on real-world dataset from modern online advertisement platform will indeed benefit the scientific community and promote the development of online bidding algorithms.

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

Received 14 October 2024; revised ; accepted

