# OpenReview forum: "BAT: Benchmark for Auto-bidding Task"
_ACM.org/TheWebConf/2025/Conference — WWW 2025 Poster_

### Official Review · Reviewer_WGHS · 2024-11-04

**Novelty:** 2
**Technical Quality:** 2

**Review:**

The paper proposes a very useful benchmark for auto-bidding tasks containing a valuable dataset. This dataset can help future research on auto-bidding a lot. However, the authors didn't release the dataset which makes it hard to comment on the value of the work. Without access to the dataset, the contents of the paper only compare the performances of some previous algorithms lacking innovations. Moreover, this paper seems to fit the Resource Track better rather than submitting to Research Tracks.

**Questions:**

In Line 569, is it possible to use methods similar to pacing, i.e. keeping the CPC around the mean, to solve this problem and achieve low loss?

**Reviewer Confidence:**

2: The reviewer is willing to defend the evaluation, but it is likely that the reviewer did not understand parts of the paper

**Scope:**

3: The work is somewhat relevant to the Web and to the track, and is of narrow interest to a sub-community

---

### Official Review · Reviewer_fJpc · 2024-11-21

**Novelty:** 3
**Technical Quality:** 3

**Review:**

The paper introduces the BAT dataset as a benchmark for evaluating auto-bidding algorithms in online advertising. It is well-structured and provides comprehensive details about the dataset, including its composition (VCG and FP auctions), filtering processes, and baselines. This BAT dataset is extensive, encompassing a diverse range of campaigns and auctions, and its design closely mimics real-world scenarios.

**Questions:**

Textual and Structural Clarity

Q1: In Line 295, there is a grammatical issue with “component od dataset.”
Q2: In Line 311, is “dow” a typo for “day”?
Q3: In Line 713, there is a missing colon after “we use the ratio.”

Figures and Visual Representations

Q4: In Figure 2, the features are not clearly highlighted. Could you provide a more detailed explanation of how these features impact algorithm performance and offer clearer visual distinctions?
Q5: In Figure 4, the x-axis labels are too small, making the figure hard to interpret. Would you consider using larger, more legible labels or an alternative visualization approach?
Q6: The logical category distribution in Figure 4 is somewhat ambiguous. Could you provide a more detailed analysis, such as exploring the relationships between logical categories, CTR, and bidding budgets?

Dataset Scope and Diversity

Q7: Does the dataset include a sufficiently wide range of industries and advertisement types? For example, have the behavioral differences across industries like automotive, food, and tech been adequately captured?
Q8: Are there clear distinctions between long-term and short-term advertising campaigns in the dataset?
Q9: Are there plans to expand the dataset to include additional metrics, such as user engagement or post-click behaviors, to offer a more comprehensive evaluation of bidding strategies?
Q10: How representative is the dataset of real-world scenarios across diverse geographic regions or platforms? Are there biases in the dataset (e.g., dominance by certain regions or categories) that might impact generalizability?

**Reviewer Confidence:**

2: The reviewer is willing to defend the evaluation, but it is likely that the reviewer did not understand parts of the paper

**Scope:**

4: The work is relevant to the Web and to the track, and is of broad interest to the community

---

### Official Review · Reviewer_XMis · 2024-11-25

**Novelty:** 3
**Technical Quality:** 6

**Review:**

Motivated by the lack of high-quality, publicly-available datasets in the autobidding domain, this submission introduces BAT (Benchmark for Auto-bidding Task), a comprehensive dataset and benchmark for evaluating and developing algorithms for real-time bidding in online advertising auctions. The dataset includes data from 10 million first-price auctions and 1 million VCG auctions, collected over several months in 2024 on a “major platform”. It contains metrics like auction outcomes, traffic data, click-through rates, and conversion rates.

Several baseline algorithms are also included, such as adaptive linear model, traffic-aware PID, model predictive PID, Mystique, and Budget-ROI, which are evaluated on budget pacing optimization, cost per click-constrained optimization, and maximizing click sums.

Strengths:

The provided dataset is comprehensive, and covers several auction types and advertiser behaviors. Benchmarks like this one also encourage reproducibility, which is good for the larger online advertising auction research community.

Weaknesses:

A minor weakness is that while the dataset contains data from first-price and VCG auctions, which are canonical in the literature, it does not contain data on any emerging auction formats beyond these two.

While the benchmark supports reinforcement learning-based approaches, the paper does not explore the abilities of reinforcement learning algorithms in real-time bidding settings. This is somewhat disappointing, as the authors have the data to do so.

**Questions:**

n/a

**Reviewer Confidence:**

3: The reviewer is confident but not certain that the evaluation is correct

**Scope:**

4: The work is relevant to the Web and to the track, and is of broad interest to the community

---

### Official Review · Reviewer_hsSq · 2024-12-02

**Novelty:** 3
**Technical Quality:** 3

**Review:**

This paper introduces a unique new dataset for evaluating automated bidding algorithms and related tasks. The dataset, collected over a month from a major platform, consists of more than 9000 advertisers, and more than 10 million records of auctions including First-Price (FP) auctions and Vickrey-Clarke-Groves (VCG) auctions. A detailed analysis of the dataset's characteristics is provided. This paper also provides two empirical auto-bidding algorithms whose performances are tested on this new dataset.

The main contribution of this paper is the dataset. As the paper mentioned, the previous dataset only contains a small number of advertisers, while this dataset contains more than 9000 advertisers. Furthermore, this dataset includes both FP and VCG auctions, while the previous dataset only contains a particular type of auction, such as the generalized second-price auction. This allows researchers to study different bidding strategies and their effectiveness under various auction mechanisms. The dataset's size also enables robust statistical analysis and training of complex machine-learning models.

However, while the proposed algorithms are effective heuristics, the paper's primary contribution appears to be the introduction of a new dataset. While this dataset is undoubtedly valuable to the whole research community. However, this makes it more suitable for a dataset track if there is any, where the emphasis is on the dataset's quality and potential impact on the research community.

**Questions:**

This paper introduces a unique new dataset for evaluating automated bidding algorithms and related tasks. The dataset, collected over a month from a major platform, consists of more than 9000 advertisers, and more than 10 million records of auctions including First-Price (FP) auctions and Vickrey-Clarke-Groves (VCG) auctions. A detailed analysis of the dataset's characteristics is provided. This paper also provides two empirical auto-bidding algorithms whose performances are tested on this new dataset.

The main contribution of this paper is the dataset. As the paper mentioned, the previous dataset only contains a small number of advertisers, while this dataset contains more than 9000 advertisers. Furthermore, this dataset includes both FP and VCG auctions, while the previous dataset only contains a particular type of auction, such as the generalized second-price auction. This allows researchers to study different bidding strategies and their effectiveness under various auction mechanisms. The dataset's size also enables robust statistical analysis and training of complex machine-learning models.

However, while the proposed algorithms are effective heuristics, the paper's primary contribution appears to be the introduction of a new dataset. While this dataset is undoubtedly valuable to the whole research community. However, this makes it more suitable for a dataset track if there is any, where the emphasis is on the dataset's quality and potential impact on the research community.

**Reviewer Confidence:**

3: The reviewer is confident but not certain that the evaluation is correct

**Scope:**

3: The work is somewhat relevant to the Web and to the track, and is of narrow interest to a sub-community

---

### Official Review · Reviewer_EMox · 2024-12-02

**Novelty:** 6
**Technical Quality:** 5

**Review:**

# Paper summary

This paper provides a dataset (Benchmark for Auto-bidding Task, and BAT for short) to support the development and evaluation of autobidding algorithms and related tasks. The authors tested the dataset on several baselines (ALM, TA-PID, M-PID, Mystique, BROI) for budget pacing/CPC constraint/click sum maximizing experiments under both VCG and first-price auction formats.

# Evaluation

This paper contributes to the rapidly expanding literature on automated bidding. Given its practical significance, it is crucial for the research community to focus not only on theoretical developments but also on the empirical evaluation of proposed algorithms. However, to the best of my knowledge, there is currently no high-quality public dataset available for research in this area. Hence, the BAT dataset in this work could be impactful if it is publicly accessible.

**Questions:**

Please clarify if the dataset or part of it would be publicly available in the future? In the manuscript, the authors only mentioned that the source code used in their analysis would be publicly available.

**Reviewer Confidence:**

2: The reviewer is willing to defend the evaluation, but it is likely that the reviewer did not understand parts of the paper

**Scope:**

4: The work is relevant to the Web and to the track, and is of broad interest to the community